# Physicians’ Words, Patients’ Response: The Role of Healthcare Counselling in Enhancing Beneficial Lifestyle Modifications for Patients with Cardiometabolic Disorders: The IACT Cross-Sectional Study

**DOI:** 10.3390/healthcare11222982

**Published:** 2023-11-18

**Authors:** Thomas Tsiampalis, Matina Kouvari, Vasiliki Belitsi, Vasiliki Kalantzi, Odysseas Androutsos, Fotini Bonoti, Demosthenes B. Panagiotakos, Rena I. Kosti

**Affiliations:** 1Department of Nutrition and Dietetics, School of Physical Education, Sports and Dietetics, University of Thessaly, 38221 Trikala, Greece or ttsiam@hua.gr (T.T.); vbelitsi@uth.gr (V.B.); vkalantzi@uth.gr (V.K.); oandroutsos@uth.gr (O.A.); fbonoti@uth.gr (F.B.); 2Department of Nutrition and Dietetics, School of Health Science and Education, Harokopio University, 17676 Athens, Greece; mkouvari@hua.gr (M.K.); dbpanag@hua.gr (D.B.P.); 3Faculty of Health, University of Canberra, Canberra, ACT 2617, Australia

**Keywords:** cardiometabolic patients, healthcare communication, lifestyle behaviors, mediterranean diet, physician recommendations, physical activity, smoking habits

## Abstract

**Background**: Preventive cardiology aims to educate patients about risk factors and the importance of mitigating them through lifestyle adjustments and medications. However, long-term adherence to recommended interventions remains a significant challenge. This study explores how physician counselling contributes to successful behavior changes in various aspects of lifestyle. **Methods**: A cross-sectional study conducted in Greece in 2022–2023 included 1988 participants. Validated questionnaires assessed patients’ characteristics, dietary habits, and lifestyle choices. **Results**: The findings revealed that patients who received lifestyle advice from physicians demonstrated increased compliance with the Mediterranean diet and a higher involvement in physical activity. Notably, they were also less likely to be non-smokers. Importantly, physicians’ recommendations had a more pronounced association with adherence level to the Mediterranean diet compared to other lifestyle behaviors. Additionally, specific dietary components like cereal, legume, and red meat consumption were significantly associated with physicians’ guidance. **Conclusions**: This study highlights the complex relationship between patients’ cardiometabolic health, lifestyle decisions, and healthcare professionals’ guidance. The substantial influence of physicians on Mediterranean diet adherence underscores the necessity for a multidisciplinary healthcare approach. Collaborative efforts involving physicians, dietitians, and fitness experts can offer comprehensive support to patients in navigating the intricate landscape of cardiometabolic health.

## 1. Introduction

Cardiometabolic diseases, which encompass cardiovascular conditions (CVD) and metabolic disorders, present a substantial worldwide health concern [1]. The third sustainable development goal recognizes the importance of CVD by targeting a one-third reduction in premature mortality due to non-communicable diseases. To meet this target, countries have to contend with various barriers limiting their ability to improve healthcare. Estimations show that 80% of cases of CVD are preventable earlier in life. This calls for cost-effective preventive strategies in the first place [2,3,4]. Within the array of strategies aimed at addressing these conditions, lifestyle adjustments, with a particular emphasis on dietary modifications, assume a central role in bolstering cardiovascular well-being and alleviating the effects of metabolic disorders [1,5,6,7,8]. In recent years, the Mediterranean diet (MD) has gained prominence as a pivotal component in preventive cardiology, emphasizing the significance of dietary patterns in maintaining cardiovascular health [9]. The MD is characterized by a high consumption of fruits, vegetables, whole grains, legumes, and healthy fats, particularly olive oil. It is also recognized for moderate fish and poultry intake, limited red meat consumption, and the incorporation of nuts. This dietary pattern has been linked to various health benefits, including reduced risks of cardiovascular diseases and improved overall well-being [10]. 

Persistent adoption of healthy lifestyle changes remains an unmet necessity in the field of preventive cardiology, posing a limitation to the overall effectiveness of these otherwise potent interventions [11]. Motivational counseling delivered by healthcare professionals holds substantial potential for instigating enduring alterations in lifestyle, thereby mitigating the risks associated with cardiovascular disease [12]. Nonetheless, this approach is accompanied by ethical considerations that necessitate careful navigation. Notably, individuals at heightened risk of cardiovascular diseases stand to gain not only from conventional pharmacotherapy but also from physicians offering guidance on lifestyle adjustments encompassing aspects like dietary choices, weight management, and physical activity [13]. The available body of evidence underscores that the mitigation of risk does not solely hinge on superior medical and pharmacological management but equivalently on the initiation and support of lifestyle changes by physicians [14]. 

As is commonly acknowledged, the effectiveness of physician counseling regarding lifestyle modifications is profoundly contingent on proficient physician–patient communication [15]. Patients not only require precise and comprehensible directives pertaining to nutrition, exercise, and other lifestyle facets but must also feel empowered and driven to implement these transformations [16]. Competent communication cultivates understanding, trust, and robust partnerships between patients and healthcare providers, thereby augmenting patients’ self-assurance and their inclination to adhere to medical and lifestyle recommendations [17]. Nevertheless, various studies centering on physician–patient interactions have brought to light instances of patient dissatisfaction, even when healthcare providers perceive their communication as satisfactory or exemplary [18]. The quality of communication substantially influences patients’ grasp of their medical condition, their perceived capacity to enact changes, and their adherence to medical and lifestyle counsel [19,20].

Thus, the primary aim of this cross-sectional observational study was to explore (a) the extent to which physicians provide counselling on lifestyle modifications in patients with cardiometabolic disorders, (b) if there is a specific patient profile where the likelihood of lifestyle counselling is higher, and (c) the effectiveness of this counselling on successful behavioral changes. This study sought to address our understanding of how counselling from healthcare practitioner impacts the detrimental behaviors of patients with cardiometabolic diseases, ultimately contributing to improved healthcare practices and patient outcomes. 

## 2. Materials and Methods

### 2.1. Study Design and Scope

Conducted during the years 2022 and 2023, this cross-sectional observational study sought to examine how physicians’ guidance influences the lifestyle changes in patients with cardiometabolic conditions. Specifically, this study investigated the effects of physicians’ recommendations on factors such as smoking habits, adherence to the Mediterranean diet, and engagement in physical activity.

### 2.2. Setting

The research was conducted across all the 7 health administrative regions into which Greece is divided, and more specifically across the following regions: (1) Attica, (2) Aegean, (3) Epirus and West Macedonia, (4) East Macedonia and Thrace, (5) Thessaly, (6) Peloponnese and West Greece, and (7) Crete. Apart from these regions, participants were also recruited from the following medical facilities: 1st Multipurpose Municipal Clinic of Solonos Athens, 2nd Multipurpose Municipal Clinic of Neos Kosmos, 3rd Municipal Clinic of Petralona, 4th Municipal Clinic of Kolonou, 6th Multipurpose Municipal Clinic of Kypseli, Trikala–Farkadona Medical Center, Pyli Medical Center, and Kalambaka Medical Center. The selection of medical facilities for participant recruitment was based on a combination of geographical representation and accessibility, aiming to ensure diverse and comprehensive coverage. The chosen facilities included both urban and regional clinics, ensuring a broad cross-section of the population. 

### 2.3. Sample

This cross-sectional study involved the enrolment of 1988 participants who had been diagnosed with CVD (specifically, coronary heart disease and stroke) or other cardiometabolic conditions such as hypertension, type 2 diabetes, type 1 diabetes, hypercholesterolemia, elevated triglycerides, obesity, and non-alcoholic fatty liver disease, out of which 1180 were female. These individuals were recruited from the specified regions using a systematic sampling approach, where participants meeting the eligibility criteria were consecutively enrolled until the desired sample size was achieved. The determination of the sample size was guided by ethical considerations and practical limitations. 

### 2.4. Eligibility Criteria

Between July 2022 and April 2023, this cross-sectional study recruited participants from the specified health regions and medical centers. To identify potential participants, the study adhered to the eligibility criteria outlined by Belitsi et al. (2023) [21]. More specifically, the following inclusion criteria were applied to identify potential participants:

Age: This study included individuals aged 18 years and older.

Diagnosis: Eligible participants were defined as those who had received a medical diagnosis of cardiovascular disease (specifically, coronary heart disease and stroke) or other cardiometabolic conditions, including hypertension, type 2 diabetes, type 1 diabetes, hypercholesterolemia, elevated triglycerides, obesity, and non-alcoholic fatty liver disease.

Treatment duration: Inclusion criteria allowed for participants who had been prescribed one or more medications for cardiometabolic disorders and had consistently taken them for at least one year.

### 2.5. Bioethics

Before initiating the study, the researchers obtained approval from the pertinent division of the Greek Ministry of Health and adhered to the principles delineated in the Declaration of Helsinki. Moreover, the study was executed in accordance with the ethical guidelines established by the University of Thessaly Ethics Committee (Ethics Approval No: 11-14/07/2022). Comprehensive information regarding the research aims and procedures was communicated to all participants, and written consent was obtained from patients before their inclusion in the study.

### 2.6. Measurements

The IAATQ-CMD questionnaire, previously established as possessing high reliability and consistency, consisted of queries pertaining to demographic and behavioral characteristics, an extensive medical history encompassing cardiovascular risk factors, and the dietary and lifestyle practices of the participants (Belitsi et al., 2023) [22]. 

#### 2.6.1. Lifestyle Counselling

Regarding the lifestyle guidance provided by physicians to patients, individuals were queried about whether their physician had furnished them with explicit lifestyle recommendations in response to their medical condition. More specifically, patients were asked whether they had been given clear recommendations by their doctor regarding their lifestyle due to their illness (yes/no). 

#### 2.6.2. Dietary and Lifestyle Characteristics

Assessment of dietary patterns was conducted using a semi-quantitative, validated, and consistently reproducible food-frequency questionnaire. Adherence to the Mediterranean diet was quantified using the MedDietScore, a scale ranging from 0 to 55 [23]. Higher scores on this scale indicated a stronger adherence to the Mediterranean diet. Patients were categorized into low or high adherers to the Mediterranean diet based on the median value of 31 units on the scale. Regarding lifestyle factors, participants were queried about their physical activity level and smoking habits. Specifically, patients were required to indicate their exercise status (yes/no), provide details about the frequency and type of exercise, and disclose their smoking status, including current smoking, past smoking, and passive smoking (all responses in yes/no format).

#### 2.6.3. Socio-Demographic, Anthropometric, and Clinical Characteristics

Participants’ socio-demographic characteristics were assessed in terms of their education level, occupational status, income, and demographic information, including marital status, age, gender, and nationality. In addition, patients’ self-reported height and weight were also recorded, while a comprehensive questionnaire was also employed in order to gather the patients’ medical history data. Further information and details about these measurements can be found elsewhere (Belitsi et al., 2023) [22]. 

To provide more information, we assessed participants’ education level, which ranged from primary school to PhD and beyond, based on which individuals were classified into the following three groups: Group I for primary school education, Group II for secondary education, and Group III for higher tertiary education. Additionally, based on their occupational status, participants were classified as: employed/freelance (Group I) and unemployed/retired (Group II), while based on their income, individuals were categorized as having either low income (<EUR 18,000/year) or at least moderate income (≥EUR 18,000/year), with the latter being in line with the OECD’s threshold for the average household net-adjusted disposable income per capita in Greece, set at approximately EUR18,000 /year. Furthermore, participants’ marital status, age, gender, and nationality were also collected.

Regarding anthropometric characteristics, participants self-reported their weight and height. From these values, the body mass index (BMI) was calculated by dividing the weight in kilograms by the square of the height in meters. In accordance with established guidelines, individuals with a BMI exceeding 29.9 kg/m^2^ were classified as obese. Furthermore, comprehensive medical history data were collected using a detailed questionnaire, which covered a range of cardiovascular risk factors and pre-existing health conditions. Further detail can be found in (Belitsi et al., 2023) [22]. 

### 2.7. Statistical Analysis

Categorical characteristics of the patients are presented as both absolute frequencies (N) and relative frequencies (%), while continuous characteristics are displayed as mean values accompanied by their corresponding standard deviations (SD). The distribution normality of continuous variables was evaluated through graphical techniques (such as histograms, PP-plots, and QQ-plots) as well as the Shapiro–Wilk test.

The association between patients’ continuous characteristics and their adherence level to the Mediterranean diet (high adherence/low adherence) was examined using the independent samples *t*-test. For categorical characteristics and the relationship between physicians’ lifestyle recommendations and patients’ characteristics, the Pearson chi-square test was employed. To explore the connection between patients’ lifestyle attributes (adherence to the Mediterranean diet, smoking status, physical activity level, and compliance with dietary recommendations) and physicians’ lifestyle recommendations, a multivariable binomial logistic regression analysis was conducted. The outcomes are presented as odds ratios (OR) along with corresponding 95% confidence intervals (CI). These logistic regression models were adjusted for patients’ demographic factors (age, sex), socioeconomic factors (educational level), and clinical characteristics (number of chronic conditions). All statistical analyses were carried out using SPSS v29.0, and statistical significance (*p*-value) was considered at <0.05 for two-tailed tests.

## 3. Results

Table 1 provides an overview of the patients’ characteristics, presented both in total and stratified by their adherence to the Mediterranean diet. The findings revealed that 59.4% of the patients were female, with an average age of 63.9 (SD 13.4) years. Furthermore, a significant proportion of patients, approximately 82.4%, had attained at least a secondary level of education. In terms of their clinical profiles, hypertension emerged as the most prevalent condition (57.4%) followed by hypercholesterolemia (37%) and type II diabetes (18.6%). Moreover, as illustrated in the findings, individuals categorized as having a high adherence to the Mediterranean diet demonstrated a notably elevated level of education (*p* < 0.001), a higher rate of unemployment (*p* < 0.001), and a correspondingly higher individual/family income (*p* < 0.001). Additionally, among those with a high adherence profile, a significantly larger proportion exhibited diagnoses of hypertension (*p* < 0.001), hypercholesterolemia (*p* < 0.001), and elevated triglyceride levels (*p* = 0.041). Conversely, the prevalence of patients diagnosed with obesity, stroke, and kidney disease was markedly lower in this group (*p*-values < 0.05 for all comparisons). In terms of lifestyle, individuals adhering rigorously to the Mediterranean diet demonstrated higher levels of physical activity (*p* < 0.001), while they also exhibited a higher likelihood of having a history of smoking (*p* < 0.001) or being passive smokers (*p* < 0.001).

Table 2 provides information concerning the patients’ profiles in which the likelihood of lifestyle counselling was higher. As depicted, patients who had received unambiguous lifestyle recommendations from their healthcare providers exhibited a notably elevated level of education (*p* < 0.001) and a significantly lower unemployment rate (*p* < 0.001). Regarding their clinical attributes, this subset of patients demonstrated a reduced likelihood of suffering from type II diabetes (*p* = 0.015), elevated triglyceride levels (*p* < 0.001), and obesity (*p* = 0.012). Conversely, they exhibited a heightened likelihood of being diagnosed with coronary artery disease (*p* = 0.029) and non-alcoholic fatty liver disease (*p* = 0.047). Noteworthy patterns also emerged in their lifestyle behaviors: patients who received explicit lifestyle recommendations from their physicians demonstrated a stronger adherence to the Mediterranean diet (*p* = 0.035) and exhibited a more active physical lifestyle (*p* = 0.016). Furthermore, this group displayed a decreased likelihood of being current smokers (*p* = 0.028) or having a history of smoking (*p* = 0.006).

Table 3 depicts the outcomes of the multi-adjusted logistic regression, probing the potential correlation between physicians’ endorsements for a healthier lifestyle and patients’ engagement in favorable behaviors. As indicated, physicians’ explicit lifestyle recommendations exhibited positive links with several beneficial behaviors. Specifically, these recommendations were associated with an increased likelihood of regular exercise, a heightened adherence level to the Mediterranean diet, and a non-smoking status. In finer detail, patients who had received unequivocal lifestyle guidance from their physicians demonstrated a 34% higher likelihood of abstaining from current smoking (OR = 1.34; 95% CI = 1.13–1.42) and a 25% higher likelihood of maintaining regular physical activity (OR = 1.25; 95% CI = 1.12–1.38). Notably, the association of physicians’ recommendations with the Mediterranean diet adherence level proved to be particularly robust; patients who had received explicit recommendations were approximately twice as likely to exhibit high adherence to the Mediterranean diet in comparison to those without such guidance (OR = 1.82; 95% CI = 1.53–2.02).

Regarding patients’ dietary habits, we conducted an in-depth exploration of the link between physicians’ recommendations for healthier lifestyles and patients’ adherence to the specific nutritional guidelines for adults in Greece with a focus on the constituent elements of the Mediterranean diet. As delineated in Table 3, patients who had received unambiguous lifestyle recommendations from their physicians demonstrated an approximately 2.5-fold increase in the likelihood of aligning with the nutritional guidelines concerning the consumption of cereals (bread, rice, pasta) and potatoes (specifically, adhering to the guidance to consume from five to eight servings of a variety of grains daily, with a predominant focus on whole grains in addition to adhering to a suggested potato consumption of approximately three servings per week; OR = 2.55; 95% CI = 1.34–4.84). This same group exhibited a 52% elevated likelihood of adhering to the nutritional guidelines related to red meat consumption (limiting red meat intake to up to one serving of lean red meat weekly, with minimal processed meat; OR = 1.52; 95% CI = 1.02–2.26). Moreover, with regards to weekly legume consumption, patients receiving explicit recommendations demonstrated a remarkable threefold increase in the likelihood of guideline adherence (meeting the recommendation of at least three servings of legumes weekly; OR = 3.38; 95% CI = 2.40–4.77). Strikingly, this subset of patients exhibited significantly diminished odds of adhering to the nutritional guidelines pertaining to both weekly egg consumption (abiding by the suggestion of up to four eggs weekly, including those integrated into dishes and desserts; OR= 0.37; 95% CI = 0.23–0.57) and daily dairy product consumption (adhering to the directive of two servings of diverse dairy products daily; OR = 0.29; 95% CI = 0.20–0.40). Lastly, it is important to note that for the remaining food groups, no statistically significant association was observed between the level of adherence to nutritional guidelines and the presence of clear lifestyle recommendations from physicians.

## 4. Discussion

By investigating the interplay between these lifestyle factors and the clarity of recommendations from physicians, this study aimed to provide a deeper understanding of the intricate pathways that guide patient behaviors. The synergy between dietary patterns, physical activity levels, and smoking habits, in conjunction with the influence of healthcare provider guidance, forms a dynamic landscape that shapes the trajectory of cardiometabolic conditions. More specifically, the intricate web of relationships uncovered by the present study provides valuable insights into the interplay between lifestyle behaviors, clinical attributes, and cardiometabolic health outcomes among patients. 

In particular, this study investigated the connection between physicians’ lifestyle recommendations and patients’ behaviors. The results showed that when physicians provided clear guidance, patients were more likely to engage in beneficial behaviors, including regular exercise, adherence to the Mediterranean diet, and not smoking. This association was particularly noticeable for diet adherence, where patients with explicit recommendations were more likely to follow guidelines for consuming grains, potatoes, and lean meats while being less likely to adhere to guidelines for egg and dairy consumption. In addition, the observed associations between high adherence to the Mediterranean diet and socioeconomic factors are intriguing, as they suggest that individuals with higher educational attainment and greater financial resources may be more inclined to adopt dietary patterns that align with the health-promoting principles of the Mediterranean diet. In addition, it is noteworthy that individuals exhibiting a higher level of adherence to the Mediterranean diet showed a notably increased prevalence of hypertension, hypercholesterolemia, and elevated triglycerides. While this finding might raise some eyebrows, one possible explanation is that those with pre-existing conditions such as hypertension or dyslipidemia may be more inclined to adopt this dietary regimen, contributing to a higher rate of diagnosed cases within this subgroup. Furthermore, it is important to recognize that genetic and familial factors could also be at play, as individuals with a genetic predisposition to these conditions often exhibit heightened awareness when it comes to their dietary choices. On the top of this, this may imply that patients who are more likely to receive lifestyle counselling from physicians are those with an impaired metabolic profile. This implies that the incentive of patients to change their lifestyle and health professionals to advise on lifestyle changes is the disease and not the prevention of disease. 

Our findings seem to agree with previous research illustrating the importance of physicians’ recommendations on patients’ lifestyles. Sun et al. (2023) [25], by investigating the impact of physicians’ recommendations on healthy behaviors among individuals with hypertension, found that those who received such recommendations were more likely to engage in healthy eating and regular exercise. However, it was noted that this group also exhibited a higher prevalence of being overweight or obese, smoking, and excessive alcohol consumption. In addition, numerous studies have also shown that when healthcare providers, particularly physicians, actively engage in counselling patients about lifestyle modifications, it can result in significant improvements in their well-being [26,27]. These recommendations encompass a wide range of behaviors, including dietary changes, increased physical activity, and smoking cessation. Effective physician–patient communication, characterized by clear, empathetic, and motivational guidance, has been linked to higher patient adherence to medical and lifestyle recommendations [28]. Patients who receive such guidance not only gain a deeper understanding of their health conditions but also feel empowered to take charge of their health [29]. The literature demonstrates that physicians’ recommendations extend far beyond medical prescriptions; they serve as catalysts for meaningful lifestyle transformations that mitigate the risk of cardiometabolic diseases and enhance overall quality of life [30]. 

Moreover, a substantial body of the literature has demonstrated the pivotal role of physicians in educating and promoting healthier lifestyles among their patients [31,32]. Educational interventions have also shown efficacy in reducing risk factors, enhancing productivity, and alleviating pain and anxiety [33,34,35]. Furthermore, numerous studies have acknowledged that medical regimens and treatments stand a higher chance of efficacy when patients adhere to physicians’ guidance [36], given the pivotal role of the physician–patient relationship in enhancing patient adherence [37].

It is widely acknowledged that physicians are uniquely positioned to provide expert advice and tailored guidance that can empower patients to make informed decisions about their lifestyles [38]. By delivering personalized recommendations, physicians not only enhance patients’ awareness of healthier habits but also provide them with tangible strategies to implement these changes. Moreover, physicians’ recommendations serve as a trusted source of information for patients, often carrying more weight than general health advice. When healthcare providers emphasize the importance of specific lifestyle modifications, patients are more likely to take them seriously and integrate them into their daily routines [39,40].

### Importance of Physicians’ Recommendations on Adherence Level to the Mediterranean Diet

The greater association of physicians’ recommendations with adherence level to the Mediterranean diet compared to physical activity and smoking habits could be explained by several factors. Patients might view dietary recommendations from their physicians as more authoritative and directly related to their overall health and well-being [41]. They might perceive dietary guidance as crucial for managing their conditions and reducing health risks, leading them to adhere more closely to the Mediterranean diet [42]. In addition, dietary recommendations, especially those related to the Mediterranean diet, often provide specific guidelines on what foods to consume and avoid. This clear and tangible guidance might make it easier for patients to make immediate changes in their eating habits, unlike the more abstract recommendations related to physical activity or smoking cessation [43]. Despite the fact that physical activity recommendations can also be clear, it is probably just harder to implement them more regularly because they require more time and commitment, while recommendations to quit smoking alone, which is a strong addiction, are often insufficient. 

Furthermore, adhering to the Mediterranean diet can align with cultural norms and preferences in certain regions [44]. The diet may include foods that are commonly consumed and enjoyed by patients, making it more feasible and appealing for them to adopt. On the other hand, physical activity and smoking habits can be deeply ingrained in lifestyle choices that are harder to change [45]. In addition, it should be noted that the Mediterranean diet is associated with a wide range of health benefits, including improved heart health, weight management, and reduced risk of chronic diseases [46]. Patients might be more motivated to adhere to a diet that offers such immediate health advantages compared to changing physical activity or quitting smoking, which may have more gradual and long-term benefits [47]. Also, during medical consultations, physicians might spend more time discussing and emphasizing dietary recommendations due to their perceived significance [48]. This extended focus on dietary advice could contribute to patients’ heightened awareness and commitment to following the Mediterranean diet [49].

## 5. Limitations

While this study contributes significantly to our comprehension of the intricate relationships between physician recommendations, patient behaviors, and adherence to cardiometabolic health guidelines, it is essential to acknowledge several limitations. The cross-sectional design of the study inherently restricted our ability to establish causal relationships, and the reliance on self-reported data introduced potential response biases. Furthermore, the study’s focus on a specific geographical context may constrain the generalizability of its findings to more diverse populations. 

## 6. Conclusions

Our findings underscore the pivotal role that physicians play in shaping patients’ lifestyle choices and behaviors. This highlights the need for a comprehensive healthcare approach, emphasizing the importance of collaboration among various healthcare professionals. Recognizing the multifaceted nature of cardiometabolic health and adopting a team-based approach are essential for equipping patients with the knowledge and resources needed for sustained lifestyle improvements. Implementing public health strategies, including enhancing physician training, promoting the Mediterranean diet, supporting smoking cessation, and encouraging physical activity, can lead to significant reductions in the prevalence of cardiometabolic conditions like hypertension and hypercholesterolemia. Ultimately, these measures can enhance the overall health and well-being of the population.

## Figures and Tables

**Table 1 healthcare-11-02982-t001:** Demographic, clinical, and lifestyle characteristics of cardiometabolic patients, both in total as well as stratified by their level of adherence to the Mediterranean diet.

		Level of Adherence to Mediterranean Diet		
	Total Sample (N = 1988)	Low Adherence (N = 912)	High Adherence (N = 1076)	*p*-Value
**Demographic characteristics**				
**Sex** (N (%) female)	1180 (59.4)	536 (58.8)	644 (59.9)	0.625
**Age** (in years; mean (SD))	63.9 (13.4)	62.8 (13.3)	64.8 (13.4)	<0.001
**Level of education** (N (%))				<0.001
Primary education	351 (17.7)	153 (16.8)	198 (18.4)	
Secondary education	985 (49.6)	497 (54.5)	488 (45.4)	
Higher-tertiary education	651 (32.8)	262 (28.7)	389 (36.2)	
**Occupational status** (N (%))				<0.001
Employed/freelance	1225 (61.6)	621 (68.1)	604 (56.1)	
Unemployed/retired	763 (38.4)	291 (31.9)	472 (43.9)	
**Marital status** (N (%))				0.907
Married/cohabitation	1328 (66.8)	608 (66.7)	720 (66.9)	
Other (single, widowed, divorced)	660 (33.2)	304 (33.3)	356 (33.1)	
**Income** (N (%))				<0.001
Less than EUR 18,000 /year	1402 (70.5)	691 (75.8)	711 (66.1)	
More than EUR 18,000 /year	586 (29.5)	221 (24.2)	365 (33.9)	
**Clinical characteristics (N (%))**				
Hypertension	1141 (57.4)	468 (51.3)	673 (62.5)	<0.001
Type II diabetes	370 (18.6)	154 (16.9)	216 (20.1)	0.069
Type I diabetes	64 (3.2)	35 (3.8)	29 (2.7)	0.150
Hypercholesterolaemia	735 (37)	282 (30.9)	453 (42.1)	<0.001
Increased triglycerides levels	354 (17.8)	145 (15.9)	209 (19.4)	0.041
Obesity	251 (12.6)	133 (14.6)	118 (11.0)	0.016
Coronary artery disease	352 (17.7)	152 (16.7)	200 (18.6)	0.264
Stroke	126 (6.3)	74 (8.1)	52 (4.8)	0.003
Kidney disease	120 (6.0)	75 (8.2)	45 (4.2)	<0.001
Non-alcoholic fatty liver disease	115 (5.8)	56 (6.1)	59 (5.5)	0.532
**Lifestyle characteristics** **(** **N (%)** **)**				
Physical activity	1213 (61.0)	499 (54.7)	714 (66.4)	<0.001
Current smoker	629 (31.6)	282 (30.9)	347 (32.2)	0.526
Past smoker	363 (26.7)	123 (19.5)	240 (32.9)	<0.001
Passive smoking	662 (48.7)	217 (34.4)	445 (61.1)	<0.001

Notes: *p*-value was based on the Pearson chi-square test (in case of categorical characteristics) and on the independent samples *t*-test (in case of continuous characteristics); level of adherence to the Mediterranean diet was measured via the MedDietScore scale, and patients were classified as low and high adherers based on the median value of the scale (median = 31 units).

**Table 2 healthcare-11-02982-t002:** Demographic, clinical, and lifestyle characteristics of cardiometabolic patients based on whether they had been given clear recommendations by their doctor regarding their lifestyle due to their illness or not.

	Have You Been Given Clear Recommendations by Your Doctor Regarding Your Lifestyle Due to Your Illness?		
	No (N= 145)	Yes (N= 1843)	*p*-Value
**Demographic characteristics**			
**Sex** (N (%))			0.490
Male	55 (37.9)	753 (40.9)	
Female	90 (62.1)	1090 (59.1)	
**Educational level** (N (%))			<0.001
Primary education	44 (30.3)	307 (16.7)	
Secondary education	54 (37.2)	931 (50.5)	
Higher-tertiary education	47 (32.4)	604 (32.8)	
**Occupational status** (N (%))			<0.001
Employed/freelance	66 (45.5)	1159 (62.9)	
Unemployed/retired	79 (54.5)	684 (37.1)	
**Marital status** (N (%))			0.980
Married/cohabitation	97 (66.9)	1231 (66.8)	
Other (single, widowed, divorced)	48 (33.1)	612 (33.2)	
**Income level** (N (%))			0.236
Less than EUR 18,000 /year	96 (66.2)	1306 (70.9)	
More than EUR 18,000 /year	49 (33.8)	537 (29.1)	
**Clinical characteristics (N (%))**			
**Hypertension**	83 (57.2)	1058 (57.4)	0.969
**Type II diabetes**	38 (26.2)	332 (18)	0.015
**Type I diabetes**	4 (2.8)	60 (3.3)	0.744
**Hypercholesterolaemia**	61 (42.1)	674 (36.6)	0.187
**Increased triglycerides levels**	58 (40)	296 (16.1)	<0.001
**Obesity**	28 (19.3)	223 (12.1)	0.012
**Coronary artery disease**	16 (11)	336 (18.2)	0.029
**Stroke**	4 (2.8)	122 (6.6)	0.066
**Kidney** **d** **isease**	8 (5.5)	112 (6.1)	0.785
**Non-alcoholic fatty liver disease**	3 (2.1)	112 (6.1)	0.047
**Lifestyle characteristics (N (%))**			
**Level of adherence to Mediterranean diet—High**	77 (53.1)	1183 (64.2)	0.035
**Physical activity—Yes**	91 (62.8)	1251 (67.9)	0.016
**Curent s** **moking status—No**	105 (72.4)	1436 (77.9)	0.028
**Past smoking status—No**	65 (61.9)	931 (74.2)	0.006
**Passive smoking—No**	49 (46.7)	647 (51.6)	0.328

Notes: *p*-value was based on the Pearson chi-square test; level of adherence to the Mediterranean diet was measured via the MedDietScore scale, and patients were classified as low and high adherers based on the median value of the scale (median = 31 units).

**Table 3 healthcare-11-02982-t003:** Odds ratio (OR) and 95% confidence interval (CI) evaluating the association between the doctors’ recommendations (main independent variable) and the patients’ lifestyle habits (Model 1–Model 11).

	Main Independent Variable: Have You Been Given Clear Recommendations by Your Doctor Regarding Your Lifestyle Due to Your Illness? (Yes Vs. No)	Odds Ratio (OR)	95% Confidence Interval (95% CI)
*Model 1*	**Dependent Variable:** Current smoking (no vs. yes)	1.34	1.13–1.42
*Model 2*	**Dependent Variable:** Physical activity (yes vs. no)	1.25	1.12–1.38
*Model 3*	**Dependent Variable:** Level of adherence to Mediterranean diet (high vs. low)	1.82	1.52–2.02
*Model 4*	**Dependent Variable:** Adhering to dietary recommendations for vegetables—*eat four servings of a variety of vegetables each day (yes vs. no)*	1.51	0.53–4.31
*Model 5*	**Dependent Variable:** Adhering to dietary recommendations for fruits—*eat three servings of a variety of fruit each day (yes vs. no)*	1.39	0.81–2.38
*Model 6*	**Dependent Variable:** Adhering to dietary recommendations for cereals (bread, rice, pasta) and potatoes—*consume from five to eight servings of a variety of grains per day. The largest amount of these should be wholemeal. Of these servings, potato consumption should be about three servings per week (yes vs. no)*	2.55	1.34–4.84
*Model 7*	**Dependent Variable:** Adhering to dietary recommendations for red meat—*consume up to one serving of lean red meat per week. Of this, as little as possible should be processed (yes vs. no)*	1.52	1.02–2.26
*Model 8*	**Dependent Variable:** Adhering to dietary recommendations for legumes—*consume at least three servings of legumes per week (yes vs. no)*	3.38	2.40–4.77
*Model 9*	**Dependent Variable:** Adhering to dietary recommendations for fish—*eat from two to three servings of a variety of fish and seafood per week. Make sure at least half of your servings are oily fish (e.g., sardines, anchovies, flounder, pollock, flounder), which are high in omega-3 fats (yes vs. no)*	1.08	0.69–1.71
*Model 10*	**Dependent Variable:** Adhering to dietary recommendations for eggs—*consume up to four eggs per week, including those used to make food and desserts (yes vs. no)*	0.37	0.23–0.57
*Model 11*	**Dependent Variable:** Adhering to dietary recommendations for dairy products–*consume two servings of a variety of dairy products per day. Prefer semi-skimmed milk and yogurt (1.5–2% fat) and low-fat cheeses (yes vs. no)*	0.29	0.20–0.40

Notes: Model 1 evaluated the association between doctor’s recommendations and patients’ current smoking status; Model 2 evaluated the association between doctor’s recommendations and patients’ physical activity status; Model 3 evaluated the association between doctor’s recommendations and patients’ level of adherence to the Mediterranean diet; Model 4 evaluated the association between doctor’s recommendations and patients’ adherence to dietary recommendations for vegetables; Model 5 evaluated the association between doctor’s recommendations and patients’ adherence to dietary recommendations for fruits; Model 6 evaluated the association between doctor’s recommendations and patients’ adherence to dietary recommendations for cereals (bread, rice, pasta) and potatoes; Model 7 evaluated the association between doctor’s recommendations and patients’ adherence to dietary recommendations for red meat; Model 8 evaluated the association between doctor’s recommendations and patients’ adherence to dietary recommendations for legumes; Model 9 evaluated the association between doctor’s recommendations and patients’ adherence to dietary recommendations for fish; Model 10 evaluated the association between doctor’s recommendations and patients’ adherence to dietary recommendations for eggs; Model 11 evaluated the association between doctor’s recommendations and patients’ adherence to dietary recommendations for dairy products. In each of the models (Model 1 to Model 11), the outcomes were derived from multiple logistic regression analyses, accounting for demographic factors (age, sex), socioeconomic indicators (educational level), and clinical attributes (the number of chronic conditions) among cardiometabolic patients, with appropriate adjustments being made. The results are presented in such a way that they explain the association between the doctor’s recommendations and the patients’ healthy lifestyle habits. Level of adherence to the Mediterranean diet was measured via the MedDietScore scale, and patients were classified as low and high adherers based on the median value of the scale (median = 31 units). Dietary recommendations were based on the Greek nutritional guidelines for adults [24].

## Data Availability

Data are available upon reasonable request (renakosti@uth.gr).

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
