# Peer review of "Physicians’ Words, Patients’ Response: The Role of Healthcare Counselling in Enhancing Beneficial Lifestyle Modifications for Patients with Cardiometabolic Disorders: The IACT Cross-Sectional Study"

_healthcare, 2023, doi:10.3390/healthcare11222982_

Round 1

Reviewer 1 Report

Comments and Suggestions for Authors

1. Please sort keywords alphabetically.

2. The authors trice repeated time when the study was conducted (lines 82, 92, 106). Please reread your paper - too many repeatings of the same things. E.g. 

"The average age of the participants was 64 years, with a mean (SD) of

100

63.9 (13.4) years."

3. Methodology section is messy. Please reconsider and clearly describe step by step all necessary aspects of your paper.

4. Questionnaires were not described adequately.

5. Abbreviation were used inattentively.

6. "Further information and details about these 142 measurements can be found elsewhere (Belitsi et al., 2023) [20]. What does it mean? The paper must work by itself, without referring to other papers for important details of methodology.

7. Provide effect size for all analyses.

8. Long paragraphs are unwanted. The paper is hardly readable.

9. table 3 is messy. What are the dependent variables? Where are multiple predictors and covariates? What did the authors did here? Please high-quality papers or guidelines how to present results on a multiple logistic regression. Models 1-11 - what is it?

 10. Percents in Table 2 seem to be incorrect. See for instance Lifestyle characteristics:

Level of adherence to Mediterranean dietHigh

77 (53.1)

1183 (64.2)

What is it? How these number do correspond to Table 1 where 

Low adherence (N= 912)

High adherence (N= 1,076)

The paper is extremely unclear, and can not be accepted in its current form. 

Author Response

Please find our point-by-point response in the attachment

Reviewer 2 Report

Comments and Suggestions for Authors

Dear Authors,

Thank you for your manuscript.

You wrote an interesting paper that aimed to understand how counseling by health care professionals influences the behavior of patients with cardiometabolic diseases, and to investigate whether there is a specific patient profile in which the likelihood of lifestyle counseling is higher. Specifically, the study looked at the impact of physicians' recommendations on factors such as smoking habits, adherence to a Mediterranean diet and engagement in physical activity.

The authors rightly emphasize that reducing the risk of cardiovascular disease depends not only on excellent medical and pharmacological treatment, but also on physicians initiating and supporting lifestyle changes. They emphasize the importance of doctor-patient communication. The quality of communication significantly affects patients' understanding of their condition, their perceived ability to make changes, and their adherence to medical and lifestyle recommendations.

The authors showed that patients who received clear lifestyle recommendations from their doctors demonstrated stronger adherence to the Mediterranean diet and a more active physical lifestyle. Moreover, the group showed a reduced likelihood of being current smokers or having smoked in the past.

The manuscript is well written and is well organized.

Some comments and suggestions:

1. In the Materials and Methodology, the authors write that the study was conducted in 7 administrative regions of Greece, for readers from outside Greece, it would be useful to add how many total admiral regions there are in Greece, it could be added how many percent of all regions in Greece are 7 regions. It is not entirely clear how the medical facilities from which the respondents were recruited were selected?

2. In the description of the sample, the authors write that the study included 1,988 participants who were diagnosed with CVD and/or other cardiometabolic disorders or risk factors, it is worth mentioning here what risk factors were taken into account.

3) The IAATQ-CMD questionnaire would be good to add to the supplement.

4. It is surprising that as many as 62.1% of women - did not receive clear recommendations from the doctor about the lifestyle resulting from the disease, what the reasons may be. This is worth addressing in part of the discussion.

5 In the discussion, I can't quite agree with the statement "This clear and tangible guidance might make it easier for patients to make immediate changes in their eating habits, unlike the more abstract recommendations related to physical activity or smoking cessation (line 302-305).

The activity recommendations recommended by specialists are also clear, it's probably just harder to implement them more regularly because they require more time and commitment.

Smoking, on the other hand, is a strong addiction, and recommendations to quit smoking alone are often insufficient, as the authors discuss in a later section.

6. In the discussion, it would be worth referring to the rather surprising results obtained, where among those with a high adherence profile, a significantly higher percentage showed a diagnosis of hypertension, hypercholesterolemia and elevated triglycerides.

7 The conclusions are too broad, they should be rewritten. Some of the conclusions are a summary rather than conclusions.

8.The beginning of the conclusions (line 320-326) should be moved to the Limitations of the study section.

Author Response

Please find our point-by-point reponse in the attachment. 

Reviewer 3 Report

Comments and Suggestions for Authors

Article

Physicians' Words, Patients' Response: The Role of Health Care Counselling in Enhancing Beneficial Lifestyle Modifications for Patients with Cardiometabolic Disorders. The IACT Cross-Sectional Study

The authors in this study investigated the complex relationship between patients' cardiometabolic health, lifestyle decisions, and healthcare professionals' counselling.

As reducing the cardiometabolic disorders is important global public health issue, this study is important contribution.

With some limitations (not-equal distribution male/female (808/1180)), the study is good designed. The work is interested and can be accepted for publication in the Healthcare, but after revision addressing the following points. Some of the corrections needed are already made, as there are 3 versions of the manuscript. The following comments for corrections are for the version 3 of the manuscript.

Introduction section:

As Mediterranean Diet (MD) is mentioned starting from the Abstract through the hole text, comment about definition, characteristics and output from MD adherence is needed here.

Results:

Why in Table 1 only results for female participants are presented?

In the Notes for Table 3 is written “Dietary recommendations were based on the Greek Nutritional Guidelines for adults”. There is no citation for those guidelines. Cite it!

Round 2

Reviewer 1 Report

Comments and Suggestions for Authors

The paper was reconsidered, but many problems are still presented. 

Comment 4: Questionnaires were not described adequately. Response: We have added more detail in the revised manuscript.

New comment 4: This was not addressed. The authors did not provide full titles of questionnaires but provided only abbreviations. The language version used was not described. Examples of statements, etc.

"Assessment of dietary patterns was conducted using a semi-quantitative, validated, 144 and consistently reproducible food-frequency questionnaire". No information was provided on this semi-quantitative questionnaire. Dimensions of the questionnaire?

Methodology section still has problems and was not reconsidered adequately.

Comment 5: Abbreviation were used inattentively. Response: The abbreviations were corrected.

New comment: This was not addressed if even titles (!!!) of questionnaires were not deciphered.

Comment 7: Provide effect size for all analyses. Response: All the effect sizes regarding the conducted analyses are being reported in Table 3.

New comment: Effect size was not provided for analyses in Tables 1, 2 and 3. Moreover, parameters of regression models should be presented (e.g., df, R Statistic).

Comment 9: table 3 is messy. What are the dependent variables? Where are multiple predictors and covariates? What did the authors did here? Please high-quality papers or guidelines how to present results on a multiple logistic regression. Models 1-11 - what is it? Response: Table 3 was reformatted and more detail about the models were added, in order to be clearer to the readers of the manuscript.

New comment: The authors stated that they used multivariable binomial logistic regression analysis, however, it is not multivariable analysis, but just Simple Logistic Regression as there are 11 models with individual predictors and one dependent variables. If you want to use miltivariable logistic regression, you should run one analysis with all 11 predictors.

I recommend the authors to consult their analyses with a statistician.
